# Rethinking Image Editing Detection in the Era of Generative AI Revolution

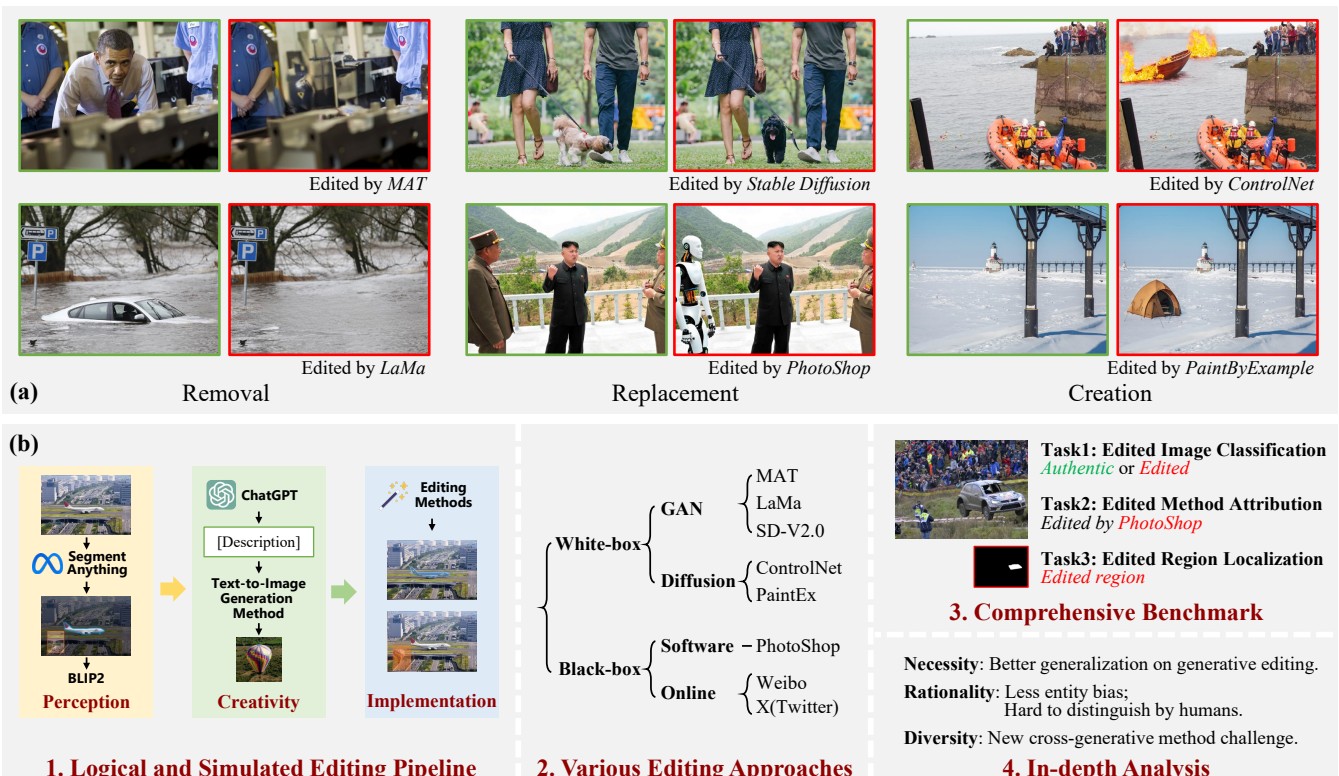

**Figure 1: GRE: a large-scale dataset and benchmark focused on the generative regional editing (manipulation) detection task. (a) Cases of edited images featuring different editing approaches and various types within the GRE dataset. (b) Illustration of several characteristics and advantages of the GRE dataset.**

## ABSTRACT

Considering that image editing and manipulation technologies pose significant threats to the authenticity and security of image content, research on image regional manipulation detection has always been a critical issue. The accelerated advancement of generative AI significantly enhances the viability and effectiveness of generative regional editing methods and has led to their gradual replacement of traditional image editing tools or algorithms. However, current research primarily focuses on traditional image tampering, and there remains a lack of a comprehensive dataset containing images edited with abundant and advanced generative regional editing methods.

We endeavor to fill this vacancy by constructing the GRE dataset, a large-scale generative regional editing detection dataset with the following advantages: 1) Integration of a logical and simulated editing pipeline, leveraging multiple large models in various modalities. 2) Inclusion of various editing approaches with distinct characteristics. 3) Provision of comprehensive benchmark and evaluation of SOTA methods across related domains. 4) Analysis of the GRE dataset from multiple dimensions including necessity, rationality, and diversity. Extensive experiments and in-depth analysis demonstrate that this larger and more comprehensive dataset will significantly enhance the development of detection methods for generative editing.

ACM MM, 2024, Melbourne, Australia
© 2024 Copyright held by the owner/author(s). Publication rights licensed to ACM.
ACM ISBN 978-x-xxxx-xxxx-x/YY/MM
https://doi.org/10.1145/nnnnnnn.nnnnnnn

## CCS CONCEPTS

• **Security and privacy → Social aspects of security and privacy**.

## KEYWORDS

Image Editing Detection, Generative Regional Editing Detection, Dataset and Benchmark

## 1 INTRODUCTION

While image editing and manipulation technologies enrich visual content, they also pose significant threats to the authenticity and security of image content in various media. Therefore, research on image regional manipulation detection has always been a critical issue. Recently, diffusion models have sparked an AI generation revolution in the field of computer vision, demonstrating remarkable performance in various task scenarios, including controllable editing [27, 28, 44, 45]. The advancement of generative technologies lowers the cost and improves the effectiveness of edits, gradually replacing traditional editing tools with generative editing methods. However, current detection researches are focused on traditional editing methods, and there remains a research gap in the detection of novel generative regional editing.

In contrast to the challenging precise control in the full image generation[1] techniques, local editing methods exhibit greater flexibility, which enables the modification of specific content in the original image [27, 40, 46], potentially altering the conveyed information. Moreover, compared to traditional manual manipulation using tools like PhotoShop, generative regional editing is more convenient and user-friendly for non-professionals, while still achieving high-quality editing results. Figure 1 (a) showcases the performance of several representative generative regional editing methods, illustrating the difficulty in distinguishing between authentic and edited images. In the present day, we can indeed assert that "Seeing is not always believing." [19] Therefore, the detection capabilities of generative regional editing merit our attention.

In this paper, we construct a novel large-scale dataset named **GRE** (Generative Regional Editing) focused on the task of detecting generative regional edits. Based on the GRE dataset, we establish a benchmark to evaluate the existing detection methods across related domains, and we analyze the dataset from multiple dimensions, including necessity, rationality, and diversity. The extensive experiments and in-depth analysis demonstrate that this larger and more comprehensive dataset will significantly enhance the development of detection methods for generative editing. Specifically, the GRE dataset offers several distinct advantages over existing related datasets, which are listed below:

*(1) Logical and Simulated Editing Pipeline.* Previously, small-scale regional editing datasets ensured logical coherence (*e.g.*, preventing the appearance of a dog in the sky) through manual manipulation, while larger datasets struggled to maintain logical consistency through a naive automated editing pipeline. To ensure logical coherence in editing, semantic richness in editing, data scale, and

---

[1]In this paper, "image generation" specifically refers to instances where all pixels are generated, while "regional editing" denotes the modification of only a portion of the pixels based on the original image. In some literature, "regional editing" is also called "manipulation."

scalability, we integrate multiple awesome large models in various modalities to construct a complete image editing pipeline including perception, creativity, and implementation.

*(2) Various Editing Approaches.* In real-world scenarios, it is impossible to know in advance the tools or methods used for editing, making it crucial to evaluate the generalization capabilities of detection models across different and even unknown editing methods. We select a variety of representative editing methods for thorough investigation. These methods vary in their architectures, including GAN-based, diffusion-based, and black-box approaches, and they also differ in their editing control mechanisms.

*(3) Comprehensive Benchmark.* Besides the binary classification task that distinguishes manipulated images from authentic ones, it is also important to improve the explainability of the image manipulation detection task in real-world media forensics scenarios by answering where and how the image is edited. We provide multi-level annotations in the dataset and propose three tasks: 1) Edited Image Classification, distinguishing whether an image is edited. 2) Edited Method Attribution, identifying the editing method used in an edited image. 3) Edited Region Localization, localizing manipulated areas within edited images. We evaluate the performance of state-of-the-art methods on these tasks, and the experiments show that the pixel-level localization task, although more challenging, is meaningful in finding edited elements within a visually rich edited image.

*(4) In-depth Analysis.* We conduct extensive experiments to analyze the key characteristics necessary for the GRE dataset to serve as a benchmark, including its necessity, rationality, and diversity. Through cross-dataset experiments with existing datasets, we validate the necessity of the GRE dataset in addressing the research gap in the detection of novel generative regional editing. TCAV analysis and user study demonstrate that the dataset exhibits no entity bias and that the editing operations are hard to distinguish by humans. Cross-editing method experiments highlight the value of the diversity of generative editing methods. These multiple dimensions collectively confirm that GRE is a high-quality dataset.

## 2 RELATED WORK

### 2.1 Generation and Manipulation Datasets

**Image Generation.** Recently, there has been a growing emphasis on the detection of generative images, leading to the introduction of numerous benchmarks such as DeepArt [36], IEEE VIP Cup [34], DE-FAKE [39], and CiFAKE [2], along with the million-scale benchmark provided by GenImage [48]. However, the generative images within these datasets are primarily suitable for image-level generation detection tasks. They do not fully meet the requirements for the edited region localization task. Creating datasets specifically for the generative regional editing detection task incurs higher costs, and its pixel-level automated editing process is more complex compared to image-level generation.

**Regional Image Editing.** Detecting tampered or edited regions in an image is a longstanding challenge. Table 1 provides a summary of scale, image source, and editing approaches of existing datasets, including Columbia [29], CASIA [5], Coverage [37], NIST16 [7], DEFACTO [20] and IMD20 [21], which are widely used and recognized. Among these datasets, only the DEFACTO dataset includes

Table 1: Summary of various regional editing detection datasets. GRE surpasses any other dataset both in scale and diversity.

| Dataset | Dataset Scale | | Original Image | | Generative Editing Approaches | | | Pipeline |
|---------|---------------|---|----------------|---|-------------------------------|---|---|----------|
|         | Edited Images | Generative Ratio(%) | Daily | News | GAN-based | Diffusion-based | Black-box | |
| Columbia[29] | 180 | 0.0 | ✓ | ✗ | ✗ | ✗ | ✗ | Random |
| CASIAv1[5] | 920 | 0.0 | ✓ | ✗ | ✗ | ✗ | ✗ | Manual |
| CASIAv2[5] | 5,063 | 0.0 | ✓ | ✗ | ✗ | ✗ | ✗ | Manual |
| Coverage[37] | 100 | 0.0 | ✓ | ✗ | ✗ | ✗ | ✗ | Manual |
| NIST16[7] | 564 | 36.9 | ✓ | ✗ | ✓ | ✗ | ✗ | Manual |
| DEFACTO[20] | 149,587 | 16.7 | ✓ | ✗ | ✓ | ✗ | ✗ | Random |
| IMD20[21] | 2,010 | 0.0 | ✓ | ✗ | ✗ | ✗ | ✗ | Manual |
| **GRE** (Ours) | 228,650 | 100.0 | ✓ | ✓ | ✓ | ✓ | ✓ | Simulated&Manual |

a relatively extensive collection of generative edited image data. Other datasets predominantly include early non-generative forms of editing (e.g., simple splice and copy-move). However, the generative editing methods employed in the DEFACTO dataset are limited, and the automated editing pipeline is relatively simple. This editing pipeline leaves noticeable traces of automation, resulting in significant generalization issues for models trained on the dataset.

## 2.2 Generative Regional Editing Methods

**Diffusion-based methods.** The emergence of diffusion models has truly propelled generative editing methods to outperform operation sequences dominated by manual interventions, both in terms of convenience and effectiveness. Stable Diffusion [27] represents an advanced text-to-image diffusion model capable. The inclusion of simple mask replacement operations during the inference process enables targeted region editing. ControlNet [46] introduces innovative modules that enable the control of pre-trained large-scale diffusion models to accommodate additional input conditions. Paint-byExample [40] explores exemplar-guided image editing rather than language-guided image editing, enabling even more precise control over the editing process.

**GAN-based methods.** However, we must also acknowledge the significant performance improvements in GAN-based image editing methods that have occurred in recent times. MAT [13] customizes an inpainting-oriented transformer block, in which the attention module aggregates non-local information exclusively from partially valid tokens, as indicated by a dynamic mask. This approach demonstrates remarkable effectiveness in addressing extensive inpainting challenges. LaMa [31] optimizes the intermediate feature maps of a network by minimizing a multi-scale consistency loss during inference. This approach adeptly handles the issue of lacking detail present at higher resolutions, resulting in improved visual quality.

## 3 GRE CONSTRUCTION

Most of the existing image generation datasets only contain full image generated samples, without considering the common scenario of regional editing within images. Most previous regional editing datasets only contain manipulation without the participation of generative models, and the creation processes lack consideration of logical rationality and semantic diversity. In contrast, our proposed GRE dataset provides various generative regional editing approaches and defines three tasks (*i.e.* edited image detection,

edited region localization, and editing method attribution) with a total of 228K images. We design an automated editing pipeline assisted by multiple large models with different modalities, capable of performing logically consistent editing operations. We compare our GRE with other public regional editing datasets, as detailed in Table 1. Overall the comparison items listed in the table, our dataset outperforms others in both scale and diversity.

## 3.1 Original Image Collection

In the context of the internet, where image content and scenes are highly complex and diverse, we select the two most frequently tampered or edited scenarios: *Daily Moment Snapshots* and *News & Public Sentiment Visuals*. In these two typical scenarios, we gather abundant original images to enhance diversity across dimensions such as scenes, content, and resolution.

*Daily Moment Snapshots* comprises user-shared pictures capturing daily life scenes and sharing moments, depicting the ordinary and personal aspects of individuals' lives. COCO [14] and Flickr2K [32] collected images from *flickr.com*, comprising photographs uploaded by amateur photographers with searchable keywords, including 40 scene categories. Similarly, DIV2K [1] and SR-RAW [47] gathered high-resolution images from a diverse set of websites and cameras, capturing snapshots of various moments and abundant contents. We select original data from these datasets, where the resolutions range from 480P to 2K. *News & Public Sentiment Visuals* include visuals intricately linked to current events, news, or public sentiment, fostering broader discussions and sparking the attention of a larger audience. VisualNews [15] is a benchmark designed for the news image caption task, consisting of a large-scale collection of news images and associated metadata. The dataset was sourced from prominent news outlets such as BBC, USA Today, and The Washington Post, among others. From this dataset, we specifically select news illustrations with resolutions exceeding 720P and possessing rich content as the original images.

## 3.2 Regional Editing Pipeline

To simulate the image editing process in real-world scenarios and ensure logical coherence in edited content, we design the editing pipelines assisted by multiple large models of different modalities, as illustrated in Figure 2. This pipeline primarily consists of three pivotal components. (1) Perception, which involves selecting the region to be edited and understanding the original image content. (2)

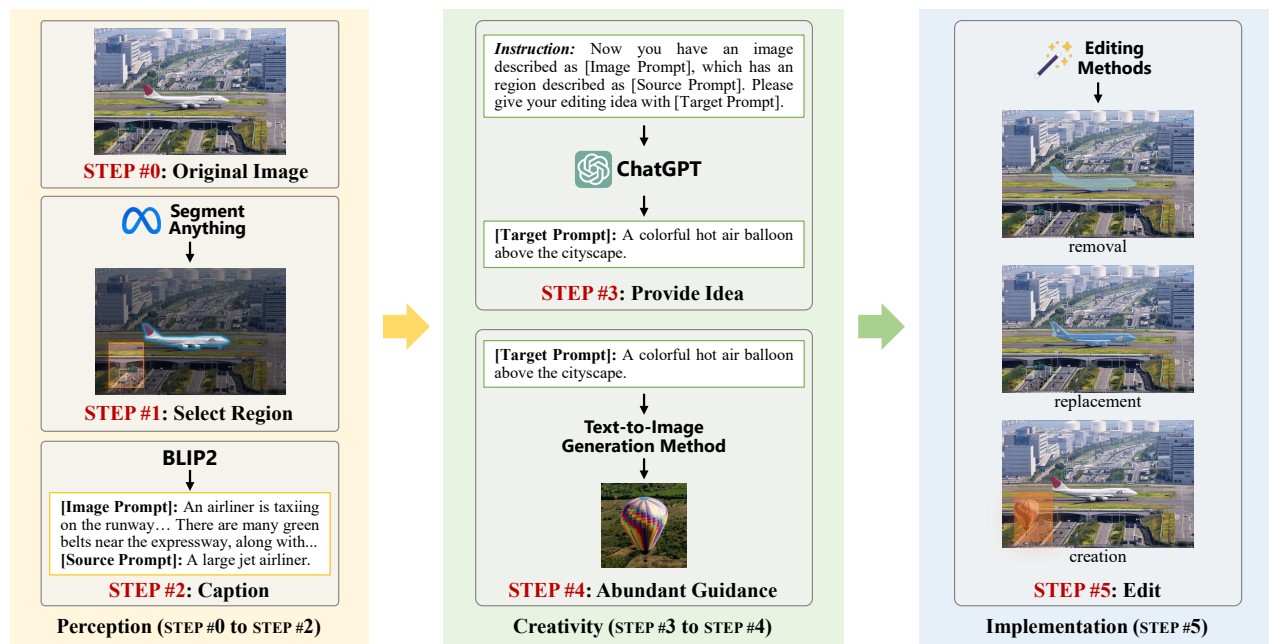

**Figure 2: Illustration of our logical and simulated pipeline with the assistance of multiple large models for regional editing.**

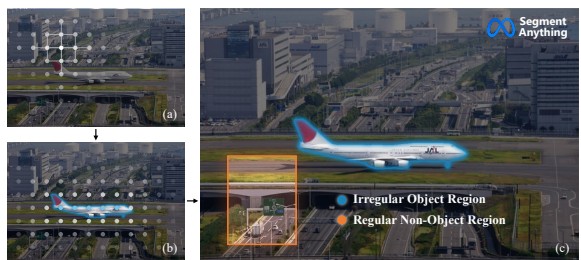

**Figure 3: Illustration of point-based SAM segmentation.**

Creativity, which involves determining the editing goal, and gathering corresponding textual descriptions and image examples (the guidance inputs for subsequent editing). (3) Implementation, which entails selecting the required guidance, employing various editing methods for multiple iterations of image editing, and filtering the optimal result.

*3.2.1 Perception.* The first crucial component of the pipeline is to achieve the perception of the original image. In this component, we aim to comprehend the image and select editing regions that are diverse and reasonable for subsequent editing. In real-world scenarios, edited regions can be broadly categorized into two types: object regions and non-object regions. For the former, editing operations such as removal or replacement can be performed, while for the latter, operations involve creating content that is not present in the original image.

To simulate the selection of objects, we employ an advanced semantic segmentation model SAM [11] to obtain precise object region masks, as illustrated in Step #1. SAM can achieve point-based segmentation. Therefore, we utilize a dense grid of points, as illustrated in Figure 3 (a), to guide SAM for multiple region predictions. For an object or region with clear semantic meaning, it should be selected by at least two points and produce similar masks. We use this

criterion to filter regions with complete semantic meaning. Conversely, outside these regions, there is a high probability of being background areas with no clear semantic meaning. In these cases, we use randomly sized rectangular regions to select these areas. We employ constraints related to size and the number of connected components to eliminate fragmented and meaningless segments. Consequently, we obtain irregular object region masks and regular non-object region masks, denoted as [Region Mask], which is the most crucial guidance for the subsequent editing process.

We employ the large-scale visual-text model BLIP2 [12] for the recognition of specified content in Step #2. We aim for BLIP2 to provide a detailed description of the original image, referred to as [Image Prompt]. Subsequently, we crop the selected region with bounding boxes enlarged by 1.3x and expect BLIP2 to provide a description of the original object or content within that region, denoted as [Source Prompt]. Finally, we analyze the coarse-grained position of the selected region in the image (using combinations such as center, top, bottom, left, and right) and incorporate this information with the [Source Prompt].

*3.2.2 Creativity.* In the real world, common editing types can be summarized as removal, replacement, and creation. Among these, removal is the most straightforward to establish, requiring only the [Region Mask] obtained in the earlier steps. However, for achieving the other editing types, the preparation of corresponding guidance that can describe the editing idea and purpose becomes essential.

ChatGPT, developed by OpenAI upon InstructGPT [23], is an excellent advisor for generating innovative editing ideas. We utilize a carefully designed instruction format to inform ChatGPT about the content of the original image and the content of the selected region for editing. We hope that it can provide diverse and realistic editing ideas that align with real-world logic in Step #3. The required text description of the editing target, [Target Prompt],

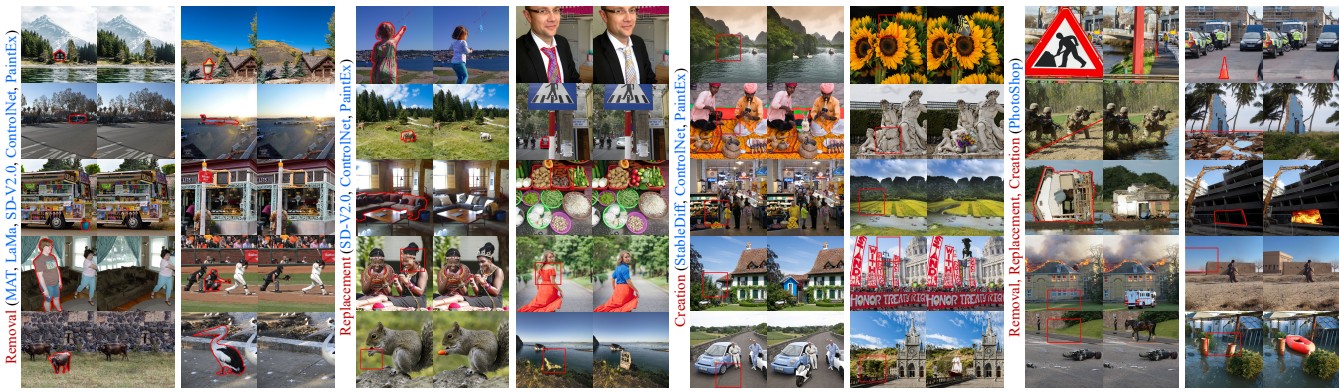

**Figure 4: Pairs of the authentic image (with edited region boundary) and corresponding edited image.**

can be extracted from its response. We leverage the currently best open-source text-to-image generation model, Stable Diffusion XL [24], to translate the text description into image examples [Target Example] in Step #4. This serves as a different form of guidance needed for the subsequent editing process. It's essential to clarify that the target examples generated in this step do not belong to the final dataset, they are merely the guidance generated by the intermediate steps.

*3.2.3 Implementation.* We have gathered comprehensive guidance information for region editing, including a precise binary mask indicating the editing region [Region Mask], textual descriptions indicating the editing target [Target Prompt], and image examples providing visual references for the editing target [Target Example]. These pieces of information offer diverse guidance for generative region editing methods, enabling end-to-end region editing.

Some works in image generation detection and attribution proposed and analyzed various generative methods from different perspectives, highlighting that different methods leave distinct traces and fingerprints [41]. Moreover, there is a noted poor generalization of detection models across data generated by different methods. To ensure diversity in edited images within our GRE dataset and to provide a reasonable benchmark for generalization evaluation, we have chosen six editing methods to complete the final component in the pipeline, implementation. These six editing methods include MAT, LaMa, Stable Diffusion V2.0 (SD-V2.0), ControlNet, PaintByExample (PaintEx), and PhotoShop, which has introduced Generative AI functionality. Details on the architecture and the required guidance for these methods, as well as other characteristics, can be found in the *Appendix*.

For each original image, we employ all white-box methods to generate corresponding edited images. However, due to the manual intervention required in the generative editing process within PhotoShop, we select only a subset of images for PhotoShop editing. When using the three diffusion models in the above-mentioned editing methods, we incorporate diverse inference steps, randomly selecting the number of steps from the set [20, 30, 50, 100] for each inference. Considering the variable quality of images generated by the diffusion-based model, multiple images are generated for each case. Subsequently, we choose the image with higher textual faithfulness based on the CLIP score [26]. Finally, we simulate real-world

scenarios by introducing perturbations to the edited images, involving random combinations of different compression algorithms and noise addition algorithms, among other post-processing operations.

### 3.3 Cases

To provide a more intuitive observation of the effectiveness of our editing pipeline, as well as the rationality and diversity of the edited images, we display cases from the dataset in Figure 4. These include three different types of edits: removal, replacement, and creation. The data are presented in pairs of authentic and edited images, with the edited region boundaries specifically marked on the authentic images. The marked regions represent the actual regions where edits occurred, meaning that changes occurred only within these regions. We also display some images manually edited using PhotoShop, which are also part of the GRE dataset.

## 4 GRE BENCHMARK

### 4.1 Benchmark Settings

*Basic Dataset Partition.* For each original image collected in GRE, we employ all white-box methods to generate corresponding edited images, resulting in a distribution from 1 (authentic) to $n-1$ (edited). Consequently, we group images edited with the same method into a subset, while all original images form the authentic subset. To ensure data uniformity and prevent data leakage, we initially partition the subset of authentic images into training, validation, and test sets in a ratio of 8 : 1 : 1. The division of each edited subset remains consistent with the authentic subset. In other words, if an original image is in the test set, all images edited from it also belong to the test set, ensuring exclusion from the training set.

*Task 1. Edited Image Classification.* This task is a 2-way image-level classification task aimed at distinguishing between authentic and edited images. We design the evaluation protocol to train models using a combination of authentic and one edited subset and then test them on other edited subsets. Specifically, we choose the SD-V2.0 subset as the training edited subset based on the experiment results presented in Table 7. This approach assesses the generalization performance of various detection methods across different types of edits. For this binary classification task, we evaluate the models using Accuracy as the performance metric.

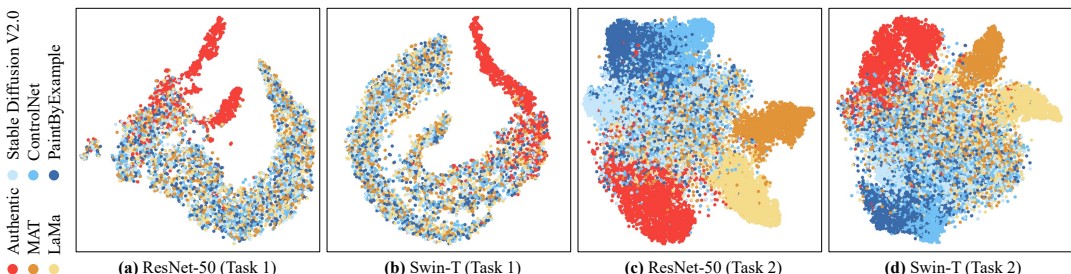

**Figure 5: The t-SNE feature visualization of the authentic images and images edited by different regional editing approaches.**

**Table 2: Comparison of related methods under the Edited Image Classification (Task 1).**

| Method | Seen Subset | | Unseen Subset | | | |
|---|---|---|---|---|---|---|
| | Authentic | SD-V2.0 | MAT | LaMa | ControlNet | PaintEx |
| ResNet-18 | 89.8 | 86.5 | 79.4 | 80.6 | 81.1 | 81.9 |
| ResNet-50 | 90.7 | 88.5 | 91.1 | **91.3** | 88.1 | 88.3 |
| DeiT-S | 91.6 | 79.3 | 72.0 | 73.8 | 71.9 | 71.5 |
| Swin-T | **95.4** | 87.8 | 85.5 | 85.6 | 86.1 | 85.2 |
| CNNSpot | 85.8 | 73.6 | 71.3 | 72.9 | 70.7 | 69.5 |
| F3Net | 82.3 | 68.1 | 62.4 | 61.7 | 59.8 | 60.5 |
| GramNet | 92.7 | **93.2** | 91.5 | 90.7 | 89.0 | 88.9 |
| Universal | 91.0 | 93.1 | **91.9** | 91.2 | **91.5** | 91.4 |

*Task 2. Edited Method Attribution.* This task refers to a $n$-way (authentic and $n − 1$ editing methods) method-level attribution task. Beyond discerning between authentic and edited images, the objective is to attribute edited images to the specific editing method employed. The evaluation protocol involves training models using all authentic and edited subsets, while the testing is performed using the basic partition of the test set. Evaluation metrics include Accuracy, F1-score, and mean Average Precision.

*Task 3. Edited Region Localization.* This task concerns a 2-way pixel-level segmentation task aimed at distinguishing between authentic and edited regions in images. For a comprehensive analysis, we introduce the protocol, which is training on a combination of the MAT subset and SD-V2.0 subset, followed by testing on other subsets. The combined training set includes one GAN-based and diffusion-based editing method respectively, a decision inspired by the experimental conclusions shown in Table 6. We use Intersection over Union (IoU) and pixel-level F1-score as assessment metrics.

## 4.2 Edited Image Classification

For a comprehensive evaluation, we provide results of several baseline models, including ResNet-18 [8], ResNet-50 [8], DeiT-S [33] and Swin-T [17]. We extend SOTA detection methods for image generation detection to the classification task of regional editing images. It is observed that the performance of GramNet [18] and Universal [22] surpasses that of CNNSpot [35], F3Net [25] and baseline. However, in Figure 5 (a) and (b), we utilize t-SNE to analyze and visualize the features of two baselines, ResNet-50 and Swin-T. An evident observation from Table 2 emerges, while the features of authentic images and edited images form a distinct classification boundary, the features of images edited using different methods do not cluster well.

**Table 3: Comparison of related methods under the Edited Method Attribution (Task 2).**

| Method | Accuracy | F1-score | mAP |
|---|---|---|---|
| ResNet-18 | 64.2 | 67.5 | 76.7 |
| ResNet-50 | 72.6 | 73.4 | 82.8 |
| Deit-S | 61.9 | 66.0 | 71.4 |
| Swin-T | **74.3** | 74.7 | 82.1 |
| DCT-CNN | 67.4 | 67.1 | 78.2 |
| DNA-Det | 72.8 | 74.5 | 82.0 |
| RepMix | 72.5 | 73.9 | **83.6** |
| POSE | 74.1 | **75.8** | 83.1 |

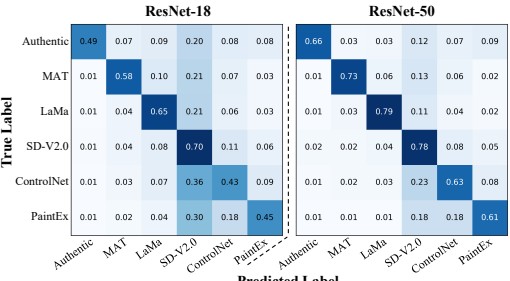

**Figure 6: Confusion matrix under the Edited Method Attribution (Task 2).**

## 4.3 Editing Method Attribution

We expand the 2-way classification labels of Task 1 to n-way attribution labels in Task 2. In addition to distinguishing between authentic and edited images, our objective is to attribute edited images to the specific editing method employed. Following the protocol, we use the authentic subset and all edited subsets for both training and testing, constituting a closed-set attribution task.

In addition to the classification baselines mentioned earlier, we also evaluate SOTA attribution models, including DCT-CNN [6], DNA-Det [42], RepMix [3], and POSE [43]. The experimental results are presented in Table 3. We also employ t-SNE to visualize the feature distributions of two baselines (ResNet-50 and Swin-T) under the protocol of Task 2, as shown in Figures 5 (c) and (d). Through comparison with Figures 5 (a) and (b), a crucial change is observed, where images edited by different methods cluster more distinctly. Additionally, various GAN-based methods can be well distinguished, while distinction among different diffusion-based methods is more challenging. Furthermore, in Figure 6, we present the confusion matrices for the attribution results of ResNet-18 and ResNet-50, aligning with the observation mentioned earlier.

**Table 4: Comparison of related methods under the Edited Region Localization (Task 3).**

| Method | Seen Subset | | Unseen Subset | | |
|---|---|---|---|---|---|
| | MAT | SD-V2.0 | LaMa | ControlNet | PaintEx |
| Unet-R50 | 72.0/80.4 | 57.9/66.1 | 29.9/38.0 | 54.7/62.9 | 62.5/70.9 |
| Unet-Eb4 | 76.3/84.7 | 65.1/74.1 | 40.5/50.7 | 60.2/69.1 | 66.5/75.5 |
| Deeplab$_{V3}$-R50 | 72.6/81.4 | 61.1/70.2 | 38.6/48.2 | 59.4/68.4 | 64.8/73.9 |
| Deeplab$_{V3}$-Eb4 | 78.1/87.9 | 59.8/69.5 | 38.4/47.6 | 54.0/64.5 | 60.4/70.6 |
| Mantra-Net* | - | - | 0.1/0.1 | 0.1/0.1 | 0.1/0.1 |
| SPAN* | - | - | 0.1/0.1 | 0.1/0.2 | 0.1/0.1 |
| PSCC-Net | 38.9/50.0 | 26.6/37.1 | 17.4/25.1 | 25.3/35.8 | 26.9/35.5 |
| MVSS-Net | 63.7/73.1 | 47.6/56.8 | 25.9/33.4 | 45.2/54.0 | 52.6/62.2 |
| SAFL-Net | 75.7/84.2 | 58.9/64.6 | 35.6/41.1 | 61.0/67.5 | 65.4/74.9 |

## 4.4 Edited Region Localization

In the context of regional edited image detection, merely distinguishing between authentic and edited images is insufficient. Locating the edited regions is a core task, and it is also the most challenging. To establish a comprehensive evaluation, we select classic baselines and representative image manipulation detection methods. We employ different combinations of classic segmentation models (Unet and Deeplab$_{V3}$) and backbones (ResNet-50 and EfficientNet-B4) as baselines for the segmentation task. For Mantra-Net [38] and SPAN [9], the core lies in their pre-trained feature extractor. Therefore, we did not retrain them on the GRE training set but rather evaluated their pre-trained models on the testing set, which is indicated by *. In addition, we evaluate MVSS-Net [4], PSCC-Net [16] and SAFL-Net [30], and the detailed experimental results are presented in Table 4.

It is worth noting that all methods exhibit acceptable localization abilities within the seen subsets. However, there is a notable lack of generalization within the unseen subsets. An important factor contributing to this phenomenon is that these methods primarily focus on non-generative forms of region editing (e.g., simple splice and copy-move). In contrast, generative regional editing approaches produce higher-quality images with less distinct boundaries for edited regions. The logic and simulated characteristics of our editing pipeline further ensure that editing operations are less perceptible. This emphasizes the value of our proposed GRE dataset for the field of regional editing detection.

## 5 GRE ANALYSIS

In this section, we conduct extensive experiments to investigate the characteristics of GRE, including its necessity, rationality, and diversity, which are essential attributes for a benchmark dataset.

### 5.1 Necessity

Existing image tampering detection datasets primarily focus on traditional types of image manipulations, such as manual edits using image editing tools like PhotoShop. Only a few datasets pay attention to manipulations performed using generative models, and the range of included generative models is very limited. Table 1 statistics some critical characteristics of current datasets. To demonstrate that existing datasets fail to effectively encompass the types of generative regional editing, as well as to highlight

**Table 5: Results of cross-dataset evaluation under the pixel-level edited region localization task.**

| Method | Training Dataset | Testing Dataset (Pixel-level F1) | | | | | |
|---|---|---|---|---|---|---|---|
| | | CASIA | DEFACTO | NIST16 | IMD20 | GRE | Avg. |
| Unet-Eb4 | CASIA | 51.8 | 19.6 | 21.4 | 19.5 | 11.0 | 24.7 |
| | DEFACTO | 5.3 | 63.2 | 4.8 | 3.7 | 2.4 | 15.9 |
| | **GRE** | 25.6 | 23.5 | 30.3 | 22.6 | 66.9 | **33.8** |
| MVSS-Net | CASIA | 44.7 | 25.1 | 26.3 | 22.2 | 16.5 | 27.0 |
| | DEFACTO | 7.9 | 54.9 | 4.3 | 4.1 | 1.7 | 14.6 |
| | **GRE** | 23.0 | 19.4 | 21.2 | 22.5 | 51.6 | **27.5** |
| SAFL-Net | CASIA | 48.2 | 15.2 | 24.0 | 21.6 | 9.8 | 23.8 |
| | DEFACTO | 6.1 | 60.5 | 4.9 | 3.0 | 2.7 | 15.4 |
| | **GRE** | 21.8 | 20.5 | 28.8 | 19.8 | 62.2 | **30.6** |

the distinctions between traditional tampering types and generative tampering types, we organize cross-dataset experiments. These experiments highlight the necessity of introducing the GRE dataset.

Among the datasets commonly used for training image tampering detection methods, we select two representative datasets: CASIA (v1&v2) [5], which contains only traditional tampering types, and DEFACTO [20], which includes traditional tampering types as well as generative tampering types implemented using GAN. In contrast, GRE encompasses tampered images edited through a variety of generative editing methods. The remaining existing datasets, due to their limited data, are used solely for testing.

We employ the best-performing models in the edited region localization task, baseline model Unet, along with two state-of-the-art methods, MVSS-Net and SAFL-Net, for cross-dataset experiments. Table 5 displays the results of cross-dataset evaluation experiments. By comparing the results of experiments using CASIA and GRE as training sets, we can elucidate the differences between traditional tampering types and generative tampering types. Although DE-FACTO includes generative tampering implemented using GAN, the experiment demonstrates that tampering performed with a single generative model does not provide sufficient generalization ability for tampering detection methods. These experiments highlight the imperative need to introduce the GRE dataset.

### 5.2 Rationality

The correlation and bias in a dataset used for training between tampered regions and specific semantic concepts can severely impair the generalization capabilities of detection methods [30]. Hence, the richness of the semantics related to the tampered regions and avoiding entity bias are critical. In the process of constructing the GRE dataset, we employ ChatGPT as the creator of editing ideas, enriching the edition semantic within the dataset and further avoiding entity bias. To further demonstrate that there is no correlation between tampered regions and specific semantic concepts in the dataset, and to validate the rationality for using ChatGPT, we use the TCAV (Testing with Concept Activation Vectors) [10], as utilized in SAFL-Net, to analyze the correlation between tampered category predictions and common semantic concepts in models trained with GRE, as shown in Figure 7.

Unet trained on CASIA and DEFACTO respectively, exhibit strong correlations between common semantic concepts and tampering detection. However, models trained on the GRE dataset

**Table 6: Results of cross-editing method evaluation under the pixel-level edited region localization task.**

| Training Subset | Testing Subset (Pixel-level IoU / F1) | | | | |
|---|---|---|---|---|---|
| | MAT | LaMa | SD-V2.0 | ControlNet | PaintEx |
| MAT | 76.1/85.0 | 27.7/36.9 | 2.8/4.4 | 7.1/10.6 | 4.4/6.7 |
| LaMa | 26.0/35.9 | 76.8/84.9 | 1.9/3.0 | 3.0/4.8 | 1.5/2.5 |
| SD-V2.0 | 15.2/21.4 | 11.2/16.2 | 57.9/67.1 | 42.5/50.5 | 53.2/62.1 |
| ControlNet | 15.2/22.3 | 5.6/8.7 | 21.8/28.2 | 70.1/78.2 | 63.9/72.9 |
| PaintEx | 13.9/19.7 | 6.0/9.0 | 33.4/41/1 | 62.1/70.2 | 76.3/84.1 |

significantly reduce this correlation. This indicates that while ensuring the richness of editing semantics, the GRE dataset successfully avoids entity bias and the correlation between tampered regions and specific semantic concepts. The situation that exists in MVSS-Net and SAFL-Net is the same but less pronounced because these methods are designed from the outset to learn semantic-agnostic features.

Additionally, a key objective in designing the entire editing pipeline is to ensure the edited images are reasonable and realistic. We conducted a user study to analyze whether the regional edited images are easily noticeable by humans. For the GRE datasets, participants could only correctly identify around 35% of the edited images, and they were confident with their wrong decisions that commonly misclassified edited images as authentic ones. Detailed procedures and results of the user study are provided in the *Appendix*, which thoroughly demonstrates the effectiveness of our designed editing pipeline and the rationality of the GRE dataset.

## 5.3 Diversity

As the category of generative editing methods is not commonly available as prior knowledge, the generalization ability across different generative editing methods becomes an important dimension for evaluating detection models. The GRE dataset includes a variety of generative editing methods featuring different architectures, requiring different types of guidance, and serving different functions. Initially, we conduct cross-editing method evaluation experiments under the image manipulation detection task to illustrate the distinct features left by different editing methods, as shown in Table 6. In this task, the detection model is required to perform pixel-level localization of edited regions, and Unet with EfficientNet-B4 is selected as the baseline model. Images edited using the same generative editing method are defined as one subset.

Specifically, the baseline model exhibits acceptable performance within the seen subset of editing methods it was trained on. However, its generalization ability significantly decreases when tested

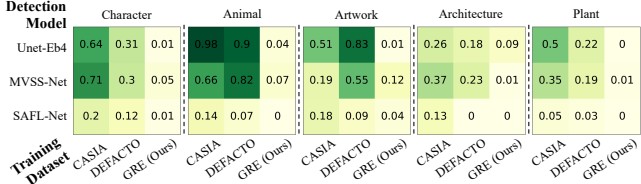

**Figure 7: Analysis of the entity bias of edited content using the TCAV.**

**Table 7: Results of cross-editing method evaluation under the image-level edited image classification task.**

| Training Subset | Testing Subset (Image-level Accuracy) | | | | | |
|---|---|---|---|---|---|---|
| | Authentic | MAT | LaMa | SD-V2.0 | ControlNet | PaintEx |
| MAT | 92.2 | 88.5 | 89.1 | 85.9 | 86.3 | 85.8 |
| LaMa | 91.9 | 89.9 | 90.0 | 87.7 | 88.1 | 87.4 |
| SD-V2.0 | 90.7 | 91.1 | 91.3 | 88.5 | 88.1 | 88.3 |
| ControlNet | 86.9 | 93.6 | 94.0 | 92.4 | 91.4 | 91.5 |
| PaintEx | 92.4 | 86.1 | 85.3 | 83.4 | 83.9 | 82.6 |

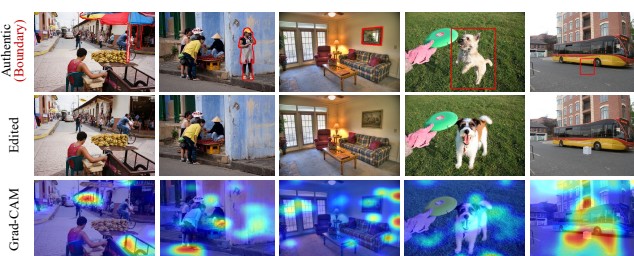

**Figure 8: Visualization of model focus regions on Edited Image Classification task using Grad-CAM.**

on unseen subsets comprising unknown editing methods. A crucial observation is that the generalization difficulty across methods with different architectures (e.g., GAN-based and diffusion-based) surpasses that between methods with the same architecture. This effectively underscores the significance and value of including a diverse range of generative editing methods in the GRE dataset.

We also conduct cross-editing method evaluation experiments under the edited image classification task, which is an image-level binary classification task determining whether an image is real or edited. We choose ResNet-50 as the baseline model and evaluated its performance across diverse editing subsets, as shown in Table 7. Notably, the baseline model exhibits commendable generalization performance when tested on unseen subsets, with no significant difference observed among different editing methods. However, further visualizations using Grad-CAM on correctly classified examples, as shown in Figure 8, reveal that the activation areas have no relation to the actual edited regions. This highlights the importance of setting the task of edited region localization and the greater challenges it presents.

## 6 CONCLUSION

In this paper, we construct a large-scale dataset and benchmark called GRE, which focuses on the task of generative regional editing detection. Unlike other existing datasets for regional editing (manipulation) detection, GRE is unique due to the diverse collection of real-world images, the simulated editing pipeline, and a variety of generative editing approaches. We introduce a benchmark composed of three crucial tasks, which provide a comprehensive evaluation of regional editing detection methods within the context of emerging scenarios. Furthermore, the in-depth analysis illustrates the necessity, rationality, and effectiveness of the GRE dataset. We plan to continue enhancing GRE by incorporating new editing methods and large models into our pipeline, to foster innovation and progress in this evolving field.

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
