# OpenReview forum: "Rethinking Image Editing Detection in the Era of Generative AI Revolution"
_acmmm.org/ACMMM/2024/Conference — MM2024 Poster_

### Official Review · Reviewer_e9gN · 2024-05-15

**Rating:** 3
**Confidence:** 3

**Summary:**

This paper presents a large-scale synthetic dataset to benchmark the performance of different generative AI models. The dataset is constructed through a sophisticated pipeline using the large-scale models SAM and ChatGPT. The authors conduct experiments to evaluate model performance on this dataset.

**Strengths:**

1. The proposed data generation pipeline is novel. This pipeline uses the large models SAM and ChatGPT to determine the target of the manipulation.
2. Although this is not the first work to benchmark forgery localisation performance on both GAN-based forgery and diffusion-based forgery, this paper extends the evaluation to more generative models and samples.

**Limitations:**

1. Huge domain gap between real forgeries. All 'creation' forgeries have rectangular manipulation areas, because the authors just splice the whole generated image without necessary post-processing. In the real world forgery, people can easily select the created foreground objects using SAM or image matting methods or Photoshop, and just splice the semantically edited objects rather than the whole image into the given pristine image to minimise the anomaly. Therefore, the real world creation forgeries may have irregular manipulation boundaries, while all the creation forgeries in the proposed dataset have perfect rectangular boundaries. As a result, the proposed dataset introduces a strong shape bias and the evaluation results are difficult to reflect the real world performance.
2. Misleading the readers. The very popular paper TruFor [1] introduces the CocoGlide benchmark for image forgery localisation, this benchmark is created with the diffusion model Glide and is widely used by the following research. However, Table 1 excludes this widely used dataset and misleads the readers by showing that none of the previous benchmarks include diffusion-based forgery.
3. Only 200 out of 228650 samples are created with human intervene, the rest (>99.9%) are pure synthetic data and 'white box' as in previous works. The highlighted contribution lies merely in the <0.1% of the proposed data.
4. A basic summary table or figure of the proposed dataset is necessary, including statistics on how many images are fogged by which method. In addition, there are many open source models for image forgery localisation, the evaluation of these open source pretrained models is mostly missing (e.g. Trufor [1] evaluated >10 models on the CocoGlide).

[1] Guillaro, Fabrizio, et al. "Trufor: Leveraging all-round clues for trustworthy image forgery detection and localization." Proceedings of the IEEE/CVF Conference on Computer Vision and Pattern Recognition. 2023.

**Suitability:**

2

---

### Official Review · Reviewer_pVPL · 2024-05-19

**Rating:** 5
**Confidence:** 4

**Summary:**

This paper proposed a novel Generative Regional Editing(GRE) dataset for Image editing detection. Specifically, a comprehensive pipeline that leverages SAM's powerful segmentation capabilities, ChatGPT's robust semantic logic performance, and the advanced functionalities of various generative models is proposed to achieve automated, high-quality region manipulation images.

Additionally, the authors designed three corresponding benchmark tasks and tested the performance of some modern methods on the GRE dataset. Extra analysis is applied to validate the superiority of the GRE dataset over previous datasets.

**Strengths:**

- The collection of tampered images for image editing detection has traditionally required extensive expert knowledge and is labor-intensive.
The authors leverage advancements in large model technologies to construct a pipeline that automates the generation of high-quality region-tampered datasets. This pipeline is innovative and addresses critical challenges in the field.
- The proposed GRE dataset fills the gap in the field of region tampering and is notably substantial in size.
- The three proposed tasks provide an excellent paradigm for effectively utilizing the GRE dataset, enabling a fair and comprehensive analysis benchmark for further study.
- The analysis of data quality is relatively comprehensive.

**Limitations:**

Although this paper is well organized, I believe there are specific details that warrant discussion and optimization.

- In line 606, the authors define Task 2 as a multi-class classification task and use the F1 score for testing. However, there are many issues with using the F1 score for multi-class classification in tampering detection. Here, I am particularly interested in a discussion about the specific use of micro-F1 or macro-F1. The authors should clarify the specific type of metric used in their paper. In previous work, such as HiFi-Net[A], the authors did not adequately distinguish this metric, which lead to unfair comparisons in subsequent research.

- The authors utilized a considerable number of evaluation algorithms and classical methods to analyze the GRE dataset, which is commendable. However, many algorithms were not properly cited, overlooking the contributions of the original developers. Please ensure that the authors give proper attention to this matter, including but not limited to:
  - line 205, TCAV [B]
  - Table 4: Unet [C] and Deeplabv3 [D]
  - line 720 EfficientNet [E]
  - line 632 and 686 t-SNE [F]
- As a paper that primarily contributes a dataset, please briefly outline your open-source plans.

## Reference
[A] Guo, X., Liu, X., Ren, Z., Grosz, S., Masi, I., & Liu, X. (2023). Hierarchical fine-grained image forgery detection and localization. In Proceedings of the IEEE/CVF Conference on Computer Vision and Pattern Recognition (pp. 3155-3165).

[B]  Kim, B., Wattenberg, M., Gilmer, J., Cai, C., Wexler, J., & Viegas, F. (2018, July). Interpretability beyond feature attribution: Quantitative testing with concept activation vectors (tcav). In International conference on machine learning (pp. 2668-2677). PMLR.

[C] Ronneberger, O., Fischer, P., & Brox, T. (2015). U-net: Convolutional networks for biomedical image segmentation. In Medical image computing and computer-assisted intervention–MICCAI 2015: 18th international conference, Munich, Germany, October 5-9, 2015, proceedings, part III 18 (pp. 234-241). Springer International Publishing.
[D] Chen, L. C., Papandreou, G., Schroff, F., & Adam, H. (2017). Rethinking atrous convolution for semantic image segmentation. arXiv preprint arXiv:1706.05587.

[E] Tan, M., & Le, Q. (2019, May). Efficientnet: Rethinking model scaling for convolutional neural networks. In International conference on machine learning (pp. 6105-6114). PMLR.

[F] Van der Maaten, L., & Hinton, G. (2008). Visualizing data using t-SNE. Journal of machine learning research, 9(11).)

**Suitability:**

3

---

### Official Review · Reviewer_G9hj · 2024-05-24

**Rating:** 4
**Confidence:** 3

**Summary:**

The passage discusses the threats posed by advanced image editing technologies to the authenticity and security of visual content. With the rise of generative AI, traditional editing methods are being replaced by generative regional editing, which is more flexible and user-friendly. However, current detection research is lagging behind these advancements. The GRE dataset is introduced to fill this gap, offering a comprehensive resource for detecting generative regional edits. It includes logical and simulated editing pipelines, various editing approaches, and comprehensive benchmarks. Extensive analysis shows that the GRE dataset significantly enhances the development of detection methods for generative editing, providing a valuable tool for researchers.

**Strengths:**

1. The passage highlights a significant gap in current research, which focuses on traditional image tampering methods, while advanced generative regional editing techniques remain under-explored. This underscores the importance and timeliness of the GRE dataset.
2. The GRE dataset is shown to significantly enhance the development of detection methods for generative editing, addressing critical issues in media forensics and contributing to the security and authenticity of visual content.

**Limitations:**

1. The passage lacks a detailed discussion on potential challenges or limitations in dataset construction, implementation, or usage. Addressing these challenges could provide a more balanced perspective on the dataset's effectiveness.
2. What is the evaluation metric in table 4?
3. While the passage mentions logical coherence in the editing pipeline, it does not provide detailed insights into how this coherence is maintained or validated. This assumption may raise concerns about the dataset's ability to reflect real-world editing scenarios accurately.

**Suitability:**

3

---

### Meta-Review · Area_Chair_piid · 2024-07-02

**Recommendation:** Accept (Poster)
**Confidence:** 5

**Metareview:**

This work received two weak accept and one weak reject for its final ratings. After carefully considering every reviewer's comments, AC decides to accept this work. However, the authors are strongly suggested to consider all suggestions in their camera ready. Specifically, the authors should try to address e9gN's concerns in the camera ready.